# Radiology Access in Rural Germany: A Nationwide Survey on Outpatient Imaging and Teleradiology

**DOI:** 10.3390/diagnostics15080962

**Published:** 2025-04-10

**Authors:** Philipp Reschke, Leon D. Gruenewald, Vitali Koch, Jennifer Gotta, Christian Booz, Scherwin Mahmoudi, Simon Bernatz, Aynur Gökduman, Elena Höhne, Katrin Eichler, Jörg Schlüchtermann, Thomas J. Vogl, Ibrahim Yel

**Affiliations:** 1Goethe University Hospital Frankfurt, 60590 Frankfurt am Main, Germany; 2Faculty of Law, Business and Economics, University of Bayreuth, 95447 Bayreuth, Germany

**Keywords:** telemedicine, teleradiology, rural health, outpatient care

## Abstract

**Background/Objectives:** This study examines the role of teleradiology and outpatient imaging in addressing gaps in rural radiology, analyzing the perspectives of referring physicians and radiologists. **Methods**: An online survey was conducted with a primary focus on evaluating the perspectives of referring physicians, including practicing surgeons, internists, and general practitioners. Their responses were then compared with those of radiologists. The survey consisted of multiple sections covering demographics, attitudes toward teleradiology, and outpatient imaging. It employed Likert scales, semantic differential scales, multiple-choice questions, and weighted ranking systems. **Results:** A total of 171 participants were included in the survey, consisting of 50 internists, 40 surgeons, 48 general practitioners, and 33 radiologists. A total of 79.2% of referring physicians rated teleradiology positively (at least 4 out of 5 stars), although its adoption in Germany remains limited, with 80.4% of referring physicians and 55.6% of radiologists reporting minimal to no use in their regions. Key concerns among referring physicians included the “lack of communication of teleradiologists with requesting physicians” (50%) and “lack of diagnostic consultations of radiologists with patients” (26.7%). In contrast, radiologists expressed significantly greater concern about technical failures (28.6% vs. 3.3%, *p* < 0.05). Additionally, 59.5% of referring physicians identified teleradiology as a crucial factor for improving access to radiology in rural areas. With regard to outpatient imaging, referring physicians valued continuity of care most highly, while radiologists prioritized minimizing wait times. **Conclusions**: Referring physicians view teleradiology and outpatient imaging as promising solutions for bridging the gap in radiology access between urban and rural areas. Addressing concerns like communication barriers and ensuring technical reliability are critical to its broader adoption and implementation.

## 1. Introduction

Access to timely and specialized healthcare remains a persistent challenge for patients in rural and remote areas [1]. This challenge not only affects patients but also places a significant burden on referring physicians, who struggle with limited access to specialized diagnostic services [2].

For instance, the ECIBC recommends biennial mammography for women aged 50–69. In most European countries women in rural areas are less likely to undergo screening compared to their urban counterparts [3]. Radiology contributes significantly to urban–rural disparities, as limited access to advanced imaging in rural areas delays diagnoses. In particular, elderly patients face additional barriers due to the difficulty of traveling long distances for radiology services, further restricting timely diagnosis and treatment [4,5,6]. Historically, radiologists have made limited efforts to serve rural communities. Dr. E. H. Skinner criticized radiologists’ failure to serve rural areas at the 1947 ACR meeting [7].

In rural Germany, patients have expressed a general preference for outpatient services provided by local physicians in practices over hospital-based care [8]. Expanding outpatient radiology can bridge urban–rural disparities by decentralizing radiology services [4].

Teleradiology might offer another solution to bridge urban–rural disparities [9,10]. Teleradiology enables certified radiologists to interpret medical images remotely, enhancing access to specialized diagnostic expertise regardless of geographic location. The foundation for modern teleradiology was laid in the 1990s with advancements in data storage and the development of Picture Archiving and Communication Systems (PACSs) [11]. With the introduction and growing adoption of CT and MR imaging, clinicians realized the need for round-the-clock radiologist expertise [12]. Teleradiology directly ensures continuous on-call coverage, particularly in rural areas where radiologist shortages are more pronounced. Despite its potential, teleradiology faces several challenges, such as data breaches from unsecured networks, unauthorized access to sensitive health information, and delays from network outages or server issues [13].

In recent years, artificial intelligence (AI) has emerged as a powerful adjunct to radiological services. Its potential is particularly relevant in underserved areas, where AI can support radiologists by automating image analysis, prioritizing critical cases, and improving diagnostic consistency [14].

The perspective of referring physicians is a critical, yet often overlooked factor in radiology, as they are the primary users of radiology reports and imaging services. Their acceptance of teleradiology and outpatient radiology care can significantly influence the effective integration of these systems into clinical workflows and patient care.

This study investigates the attitudes of referring physicians toward teleradiology and outpatient imaging and compares them with those of radiologists, emphasizing acceptance, perceived benefits, and key challenges.

## 2. Methods

Ethical approval was not required, as the survey focused solely on physicians’ opinions. The study was conducted in accordance with ethical guidelines, and all analyses complied with local data protection regulations.

### 2.1. Study Design

The study focused on practicing surgeons, internists, and general practitioners in urban and rural Germany who regularly utilize radiological services. Radiologists’ perspectives were used to compare their viewpoints as imaging providers with those of referring physicians. To ensure nationwide representation, recruitment was stratified across all 16 federal states of Germany, including metropolitan areas such as Berlin, Hamburg, and Munich and rural districts in states like Brandenburg and Saxony-Anhalt.

Eligible participants were identified through hospital and practice websites. A stratified random sampling approach was employed to achieve a balanced distribution of participants across various subspecialties and regions all across Germany. A total of 2195 physician contacts were screened, from which 453 were randomly selected for invitation in a stratified manner to reflect the proportional distribution of specialties and regional coverage in Germany. Stratification ensured a proportional representation of internists, general practitioners, surgeons, and radiologists from different regions.

Potential participants were invited through an email containing the study purpose and a survey link. Up to one reminder email was sent within a four-week period.

A pilot testing phase with a small group of 10 physicians was conducted to refine survey questions for clarity and face validity. Randomization of the question order was implemented to mitigate order effects. The survey assured anonymity.

### 2.2. Inclusion and Exclusion Criteria

Participants were included if they provided demographic data and answered at least one core question in the survey. Exclusion criteria included retirement, not routinely requesting or providing radiology services, failing to provide demographic data, or not answering any survey question.

### 2.3. Questionnaire

In this study, *teleradiology* refers to the remote interpretation of imaging studies by board-certified radiologists via digital transmission. The questionnaire focused on teleradiology modalities commonly practiced in Germany, primarily including computed tomography (CT), plain radiographs (X-rays), and magnetic resonance imaging (MRI). Ultrasound and interventional procedures were not considered relevant for teleradiology, as they typically require on-site performance and direct patient contact.

The survey included sections on demographics, attitudes toward radiology services, teleradiology use, and views on outpatient imaging. A variety of question formats were used: 5-star Likert scales assessed attitudes toward teleradiology and the impact of outpatient imaging on rural access (1 = negative/impairment and 5 = positive/improvement). Semantic differential scales (−100 to +100) evaluated the impact of outpatient shifts on wait times and trust in radiology reports (practice vs. hospital). Single-choice questions covered teleradiology adoption, concerns, benefits, and rural solutions. A 7-item ranking assessed key advantages of outpatient radiology, with weighted scoring based on rank position (Appendix A).

### 2.4. Data Analysis

The statistical analysis was conducted using MedCalc (Windows Version 20.1, MedCalc, New York, NY, USA) and Python (version 3.11). Descriptive statistics, including the mean, median, mode, and standard deviation, were calculated to summarize participant demographics and survey responses. Ranking data were analyzed with weighted scores, allowing for priority comparisons between radiologists and referring physicians. Various statistical tests, such as *t*-tests, Fisher’s Exact Test, and Chi-square, were used to perform comparative analysis between groups, with the significance set at *p* < 0.05.

## 3. Results

### 3.1. Physician Characteristics

Of the 453 physicians initially invited to participate in the study, several were excluded for specific reasons: 210 did not answer the survey, 41 did not routinely utilize or provide radiology services, 20 only completed the demographic section without responding to other survey questions, and 11 were retired.

The final study population comprised 171 participants: 50 internists, 40 surgeons, 48 general practitioners, and 33 radiologists (Figure 1).

A post hoc power analysis was conducted to evaluate whether the final sample size (*N* = 171) provided sufficient statistical power for the planned group comparisons. For a one-way ANOVA comparing satisfaction scores across four specialties (internists, general practitioners, surgeons, and radiologists), the achieved power was 0.88 to detect a medium effect size (f = 0.25) at a significance level of α = 0.05.

All general practitioners worked in practices, whereas the internists, surgeons, and radiologists were employed in hospitals. Among the 123 hospital-based physicians (internists, surgeons, and radiologists), 82 were residents, 39 were senior physicians, and 2 held the position of chief physician. The distribution of hospital-based physicians in the study closely mirrored national figures reported by the German Medical Association [15]. A Chi-square goodness-of-fit test showed no significant deviation from the expected national distribution. The final sample of 171 physicians included participants from all 16 federal states in Germany, with a balanced distribution across specialties and regions (Table 1).

The average time required to complete the survey, after excluding outliers, was 6:43 min.

Internists had an average experience of 6.9 years (mode: 6 years), surgeons averaged 11.5 years (mode: 10 years), and radiologists averaged 5 years (mode: 5 years). In contrast, general practitioners had significantly more experience, averaging 19.2 years (mode: 15 years; *p* < 0.05, one-way ANOVA test) (Table 2).

### 3.2. Attitude Towards Teleradiology

Referring physicians expressed a positive attitude towards teleradiology, with 41.6% giving it a full 5 out of 5 stars and 37.7% rating it with 4 stars. Neutral ratings (3 stars) accounted for 14.3%, while only 6.5% rated it with 2 stars (Figure 2).

Surgeons and internists rated teleradiology with an average score of 4.3 ± 0.9, with 86.4% giving 4 or 5 stars. Radiologists rated it 4.44 ± 0.68, with 88.9% awarding 4 or 5 stars. In contrast, general practitioners gave a lower average rating of 3.8 ± 0.9, with 63.6% assigning 3 or 4 stars. There was no significant difference between referring physicians and radiologists (one-way ANOVA test, *p* > 0.05).

A total of 55.6% of radiologists and 80.4% of referring physicians reported either no implementation or only limited adoption of teleradiology. However, radiologists were more likely to report moderate use (44.4% of radiologists vs. 19.6% of referring physicians; *p* < 0.05, Fisher’s Exact Test). None in either group reported widespread implementation (Figure 3).

Referring physicians expressed more concerns about communication-related issues than radiologists, such as “lack of communication of teleradiologists with requesting physicians” (50% vs. 28.6%; *p* < 0.05, Fisher’s Exact Test) and “lack of diagnostic consultations of radiologists with patients” (26.7% vs. 14.3%; *p* = 0.05, Fisher’s Exact Test).

Radiologists were significantly more concerned about technical failures (28.6% vs. 3.3%; *p* < 0.001, Fisher’s Exact Test) and expressed more concern about data protection (14.3% vs. 3.3%; *p* < 0.05, Fisher’s Exact Test) than referring physicians (Figure 4).

With regard to the benefits of teleradiology, referring physicians placed the highest value on “possibility to obtain expert opinions from subspecialized radiologists” (50.9% referring physicians vs. 30% radiologists; *p* < 0.05, Fisher’s Exact Test) and the “possibility to get second opinions when a radiology report is unclear” compared to radiologists (21.6% referring physicians vs. 10% radiologists; *p* < 0.05, Fisher’s Exact Test).

In contrast, radiologists placed more importance on “real-time image transmission for remote reporting than referring physicians” (40% radiologists vs. 17.65% referring physicians; *p* < 0.001, Fisher’s Exact Test) (Figure 5).

The most critical factor for improving radiology access in rural areas was teleradiology, cited by 59.5% of referring physicians. Additional influential factors included increasing the number of radiology practices in rural locations (27.9%) and offering radiology services within local hospitals (12.7%) (Figure 6).

### 3.3. Outpatient Care

Referring physicians rated the potential improvement of imaging access through outpatient care significantly higher than radiologists (*p* < 0.05, Chi-square test). Referring physicians assigned a 5-star rating most frequently (43.5%), followed by a 4-star rating (39.1%).

Radiologists rated it 4 stars (44.4%) and 3 stars (33.3%) most frequently. The average rating for referring physicians was 4.2 (±0.74), while for radiologists it was 3.9 (±0.9) (Figure 7).

Referring physicians perceived outpatient imaging to reduce waiting times, with an average impact score of −37.3 ± 61.6 on a scale from −100 (shorter waiting) to +100 (longer waiting time) (Figure 8).

Confidence in radiological findings was assessed on a scale from −100 (strong preference for practice-based radiology reports) to 100 (strong preference for hospital-based radiology reports). Radiologists had significantly higher confidence in hospital-based radiology reports (mean +45.56, ±29.1) than the referring physicians (mean −9.77, ±53.83, *p* < 0.0001; two-sample *t*-test with Welch’s correction), who showed a slight preference for practice-based reports (Figure 9).

The referring physicians identified the top priorities for performing routine radiology services in practices as the continuity of care through regular follow-up examinations (174 points), shorter waiting times in practices, thanks to the absence of emergency care (171 points), and cost efficiency (159 points). Other key considerations included the specialization of radiology practices in specific imaging techniques (135 points), the availability of short-term appointments (128 points), and shorter travel distances for patients (116 points) (Figure 10).

However, radiologists ranked shorter wait times the highest, followed by patient comfort and accessibility, placing continuity of care last.

## 4. Discussion

Despite advancements, the surge in imaging demand has outpaced the capacity in rural areas [16]. Access to radiological examinations in rural areas is affected by various factors, including the geographical distance to radiology facilities, a shortage of specialized personnel, and inadequate medical infrastructure. In Germany, CT and MRI scanners are less common in sparsely populated regions, leading to longer wait times for diagnostic imaging in these areas and highlighting a significant urban–rural disparity [17]. This study evaluates how outpatient radiology services and teleradiology can help address this imbalance based on the perception of referring physicians.

We report six main findings. First, teleradiology usage is limited in Germany, with 80.4% of referring physicians reporting either minimal or no use in their region (Figure 3). Despite low usage, 79.2% of referring physicians rated teleradiology positively (at least 4 out of 5 stars) (Figure 2). Second, referring physicians were more concerned about communication issues, such as “lack of communication of teleradiologists with requesting physicians” and “missing diagnostic discussions of radiologists with patients”, while radiologists expressed significantly greater concerns about “technical failures in teleradiology” (*p* < 0.05) (Figure 4). Third, referring physicians highlighted access to specialist opinions as a key benefit of teleradiology, with 50.9% emphasizing its importance in obtaining expert insights from subspecialized radiologists (Figure 5). Fourth, teleradiology was identified by 59.5% of referring physicians as the most important factor for improving radiology access in rural areas, ranking higher in importance than enhanced access to imaging in local hospitals and practices. Fifth, the majority of referring physicians indicated a positive inclination toward outpatient imaging (82.6% rated at least 4 out of 5 stars) (Figure 6). Lastly, referring physicians ranked continuity of care as the most important benefit of outpatient radiology services, whereas radiologists ranked it as the least important (Figure 8).

Although positively viewed, teleradiology remains underutilized in many regions of Germany according to the participants. By 2019, 85.6% of U.S. radiologists utilized teleradiology [18]. In Europe, teleradiology adoption has been slower than in the U.S., potentially due to stricter regulations and linguistic barriers [19]. For instance, the X-Ray Ordinance in Germany allows teleradiology under specific conditions (e.g., night and weekend services), requiring on-site technical support and direct communication with the teleradiologist (§ 3(4) RÖV; § 23(1) RÖV). Reluctance to adopt teleradiology may stem from societal concerns about data security and strict legal regulations [20,21]. However, this general societal concern is not reflected in the responses of this survey. Surveyed radiologists identified technical failures as the greatest obstacle to teleradiology, likely due to their heavy reliance on functional technology. Technical failures are not uncommon in clinical practice. The transfer of large digital image files can increase the risk of such failures [22,23]. On-site radiologists would be essential to maintain radiological services during technical difficulties, especially for emergency cases [24,25].

The main concern among referring physicians was the lack of effective communication between referring physicians and teleradiologists. This was likely due to limited interaction and the difficulty referring physicians face in reaching teleradiologists for timely communication and clarification. Communication with all parties could be crucial for referring physicians, as it might be an effective tool for ensuring accurate diagnoses and correcting misdiagnoses [26].

The second major concern was the lack of diagnostic consultations of radiologists with patients. This underscores the need for radiologists to engage with patients directly, addressing their questions [27].

In our study, most referring physicians highlighted the availability of specialist opinions as the most valued benefit of teleradiology. With the increasing subspecialization in radiology, it is becoming more challenging to ensure the widespread availability of subspecialized radiologists across all regions [28]. As a consequence, teleradiology could provide access to subspecialized radiologists, ensuring expert opinions’ availability in underserved regions. Importantly, the expected role of teleradiologists extends beyond merely delivering radiology reports. They could offer expert guidance on imaging findings and their clinical implications. Regrettably, some teleradiology companies focus solely on delivering reports. In the United States, commercial teleradiology services like NightHawk and vRad are increasingly competing with traditional on-site radiology facilities. The NightHawk model enables radiologists to work remotely, including from overseas, providing night coverage [29,30]. While these commercialized services offer on-demand, after-hours coverage from off-site locations, they are particularly noted for their high cost efficiency [30,31,32]. These radiology alternatives could carry the risk of neglecting diagnostic quality and subspecialization due to cost considerations. According to Larson and Janower in the article “The Nighthawk: Bird of Paradise or Albatross?”, the nighthawk model poses significant risks to the specialty by undermining the role of radiologists, increasing competition from non-radiologists, introducing economic and legal uncertainties, compromising quality control, and potentially leading to the outsourcing of radiology services entirely [30]. 

According to the referring physicians across all specialties, teleradiology offers the highest potential to bridge urban–rural disparities, surpassing even local access to imaging in hospitals and practices. An example confirming this preference is the STARR network in Arizona, which illustrates how teleradiology and telemedicine can effectively expand healthcare access and improve diagnostic services in rural areas. In this hub and spoke model, a central “hub” hospital provides expertise and support to smaller, rural “spoke” hospitals, enabling timely diagnoses and treatment. In this system, remote hospitals connect with Mayo Clinic’s stroke specialists, enabling timely stroke diagnoses and treatments via teleconsultation. For example, a patient at Copper Queen Community Hospital, a spoke hospital located 380 km from the Mayo Clinic hub, could receive life-saving thrombolysis within the critical window, made possible by the swift transmission of images and expert consultation from the hub [33]. This model, proven effective in Arizona, could be replicated in Germany and across Europe, where it could reduce healthcare disparities and enhance access to specialized care, not just for strokes but for other time-sensitive conditions.

Beyond improving access and diagnostic timeliness, teleradiology offers measurable economic advantages for rural healthcare systems. A well-documented analysis from rural hospitals in Oklahoma quantified these effects across four key domains: (1) cost savings from outsourcing radiologist services, (2) reduced travel expenses for patients, (3) fewer missed hours of work, and (4) increased retention of follow-up care expenditures in the local economy. For example, one hospital reduced its annual radiology costs from USD 202,000 to USD 60,000 by outsourcing 6000 reads, yielding an annual savings of USD 142,000. On the patient side, in Poteau, Oklahoma, 5% of 27,600 annual radiology cases (*n* = 1380) were classified as urgent and would have otherwise required referral to an external facility 31 miles away. By avoiding these transfers, patients saved USD 30.07 per trip in travel costs, totaling USD 41,497 annually. Teleradiology also reinforced local economies by increasing the likelihood of follow-up laboratory and pharmacy services being conducted in the community. In Poteau, the estimated annual value of these retained services reached USD 1.63 million. Cumulatively, the total economic benefit of teleradiology in this single rural hospital exceeded USD 1.8 million [34].

The positive attitude of referring physicians towards outpatient imaging was consistent with previous studies. The article by Hofmann et al. outlines how outpatient care models can alleviate pressure on hospital resources and optimize the allocation of imaging services, particularly by separating routine examinations from emergency demands. The study also shows that facilities for outpatient imaging generally have shorter wait times compared to public hospitals, which aligns with the perspective of referring physicians, evaluated in our study [35]. After all, timely access to medical imaging is essential for accurate diagnostics and prompt treatment [36].

Referring physicians especially valued the continuity of care provided by regular follow-up examinations in outpatient settings. The expectation behind this inclination might be that outpatient radiology services offer a more personalized approach to patient care compared to hospital settings, where patients often encounter different physicians during their appointments. In outpatient facilities, referring physicians and patients typically engage with the same radiologist for follow-up appointments. This continuity of care might foster trust, as the patient and the referring physician are already familiar with the radiologist. As a result, outpatient settings can provide a point of contact, ensuring that diagnostic information is conveyed clearly and promptly [37,38,39]. 

In contrast, radiologists prioritized shorter wait times, followed by patient comfort and accessibility, placing continuity of care last. This reflects their focus on operational efficiency and workflow optimization. 

Furthermore, several prior studies have shown that patients also wish to communicate directly with radiologists about the reported findings, indicating a desire for more personalized care and direct involvement in their healthcare process [40,41,42].

This study has several limitations to acknowledge. First, the relatively small sample size may not fully capture the diversity of opinions among all referring physicians in Germany. Although participants from all federal states were included using a stratified sampling strategy, the final cohort of 171 physicians represents only a small fraction of the national physician population. However, this sample size provides sufficient statistical power to detect moderate effect sizes with acceptable confidence, particularly for pooled or grouped analyses.

Second, we acknowledge that decentralizing radiology services through outpatient care can only reduce urban–rural disparities if imaging infrastructure is physically present in rural areas. In our study, we refer to outpatient imaging as a potential solution primarily when supported by local facilities.

Third, the reliance on self-reported data also poses a risk of response bias. Fourth, the study is limited to survey-based insights without incorporating actual measurements or objective assessments, which may affect the robustness of the conclusions.

To support the adoption of teleradiology in rural areas, regulations should be modernized to allow broader routine use beyond emergency settings. Funding and reimbursement incentives are needed to support rural imaging infrastructure in Germany and Europe Infrastructure efforts should prioritize expanding MRI and CT access for outpatient imaging in rural areas, along with CME-accredited training in telecommunication for both radiologists and referring physicians. To address communication barriers between referring physicians and teleradiologists, virtual reading rooms enabling real-time communication via secure platforms should be implemented. Integrating structured feedback and messaging tools into PACS/RIS systems would further support asynchronous, traceable dialogue and improve clarity in diagnostic communication.

Future research should also investigate technical safety measures and regulatory challenges to facilitate the safe and efficient integration of teleradiology and outpatient imaging within the European radiology landscape.

## 5. Conclusions

The solution to limited radiological access in rural areas could lie in expanding teleradiology and outpatient imaging services. Outpatient imaging centers reduce the burden of radiology departments in hospitals by handling routine and follow-up imaging, reducing waiting times. Teleradiology eliminates geographical barriers through remote interpretation, enabling expert review and minimizing diagnostic delays. Therefore, teleradiology and outpatient imaging might enhance access to radiological services in rural areas. Aligning the priorities of referring physicians and radiologists is crucial to fully harness these benefits for better patient care and efficiency.

## Figures and Tables

**Figure 1 diagnostics-15-00962-f001:**
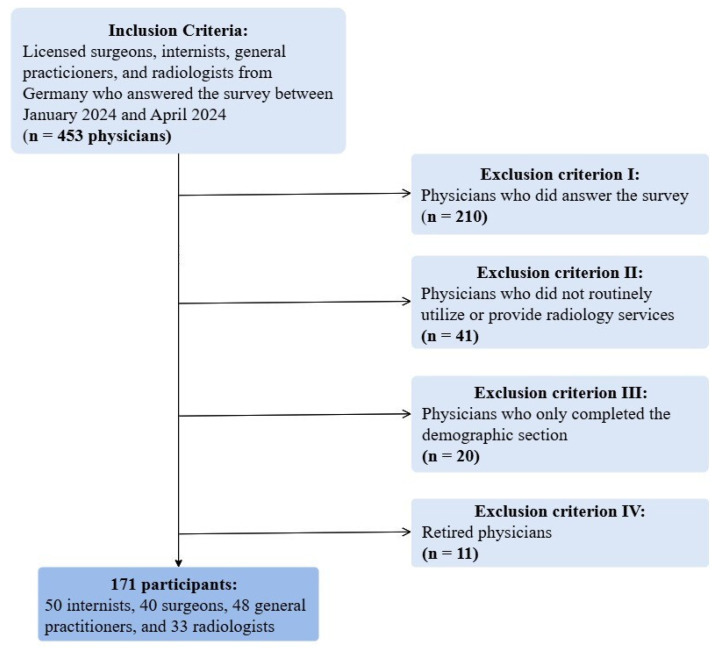
Inclusion and exclusion criteria.

**Figure 2 diagnostics-15-00962-f002:**
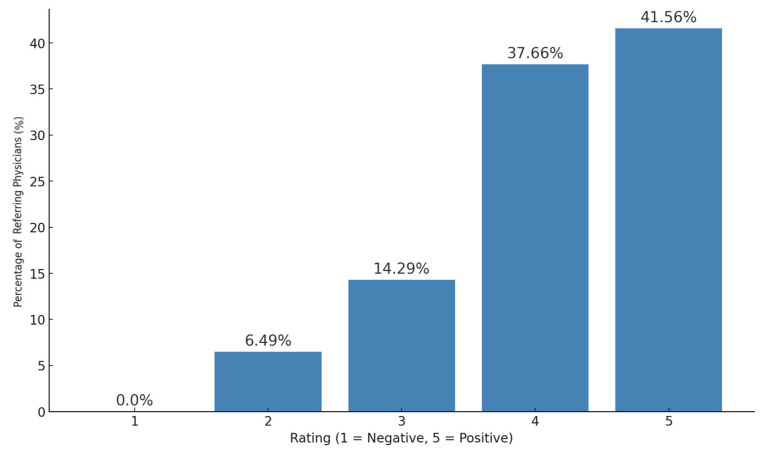
Attitude towards teleradiology.

**Figure 3 diagnostics-15-00962-f003:**
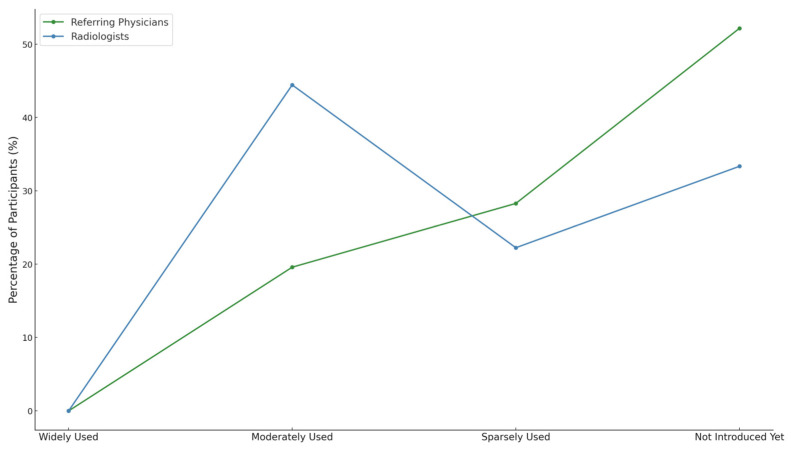
Perceptions of current usage of teleradiology.

**Figure 4 diagnostics-15-00962-f004:**
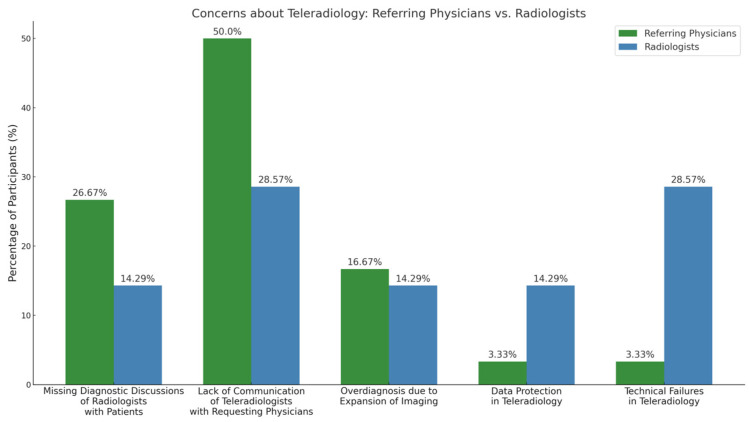
Concerns about teleradiology.

**Figure 5 diagnostics-15-00962-f005:**
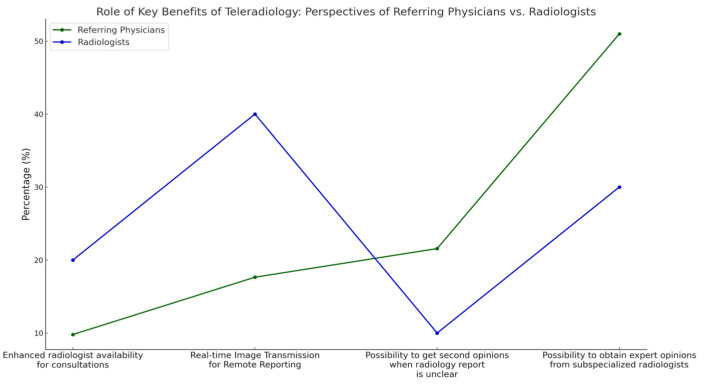
Perceived benefits of teleradiology in enhancing access to radiology examinations.

**Figure 6 diagnostics-15-00962-f006:**
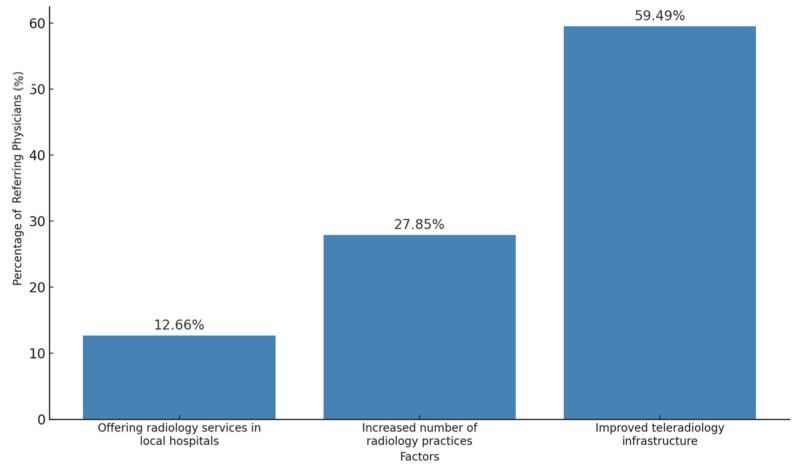
Rural imaging coverage solutions.

**Figure 7 diagnostics-15-00962-f007:**
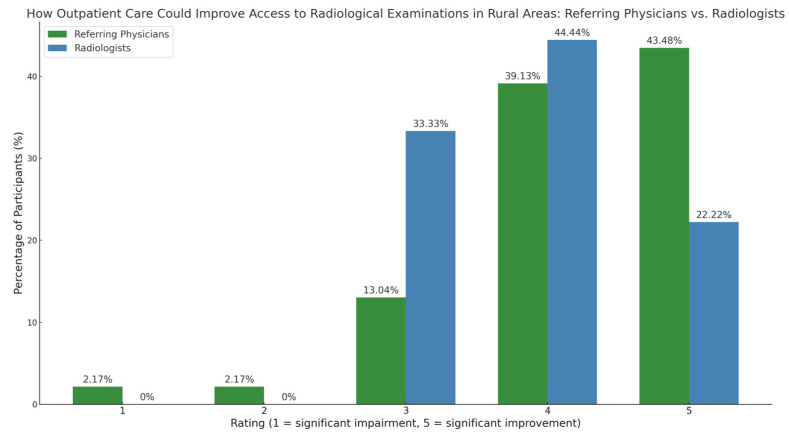
The role of outpatient imaging in improving access to radiology in rural areas.

**Figure 8 diagnostics-15-00962-f008:**
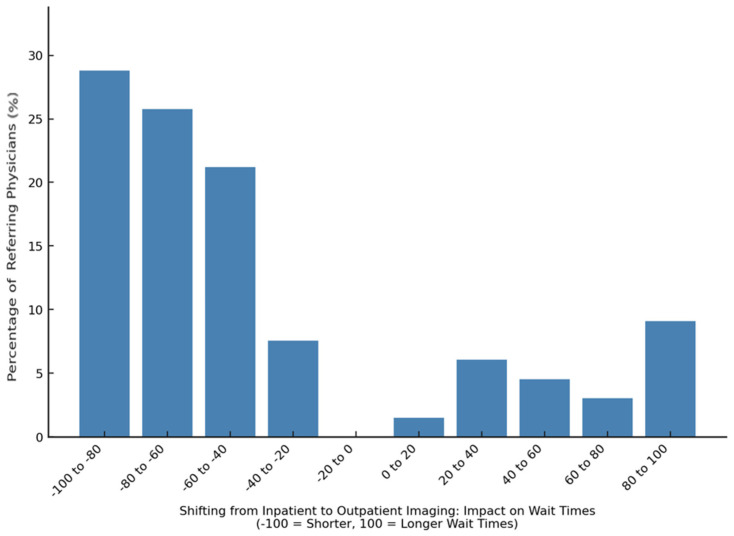
Perceived impact of outpatient radiology on waiting times among referring physicians.

**Figure 9 diagnostics-15-00962-f009:**
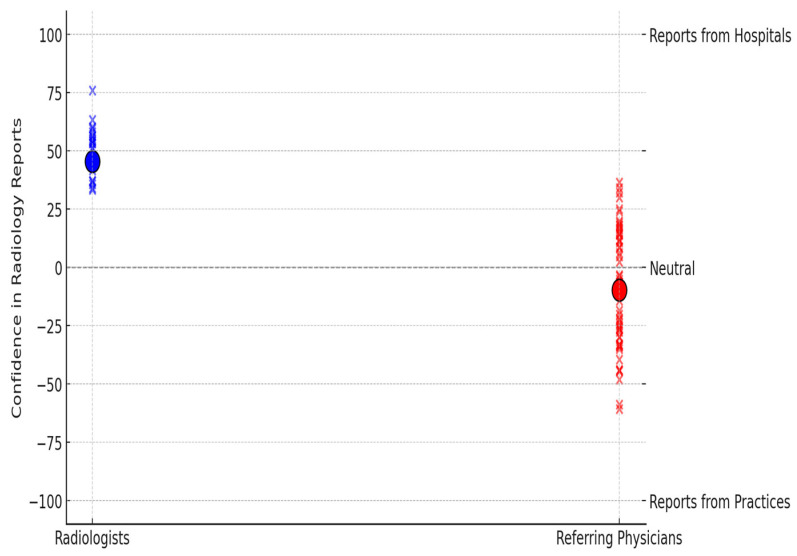
Trust in radiology reports from hospitals vs. private practices.

**Figure 10 diagnostics-15-00962-f010:**
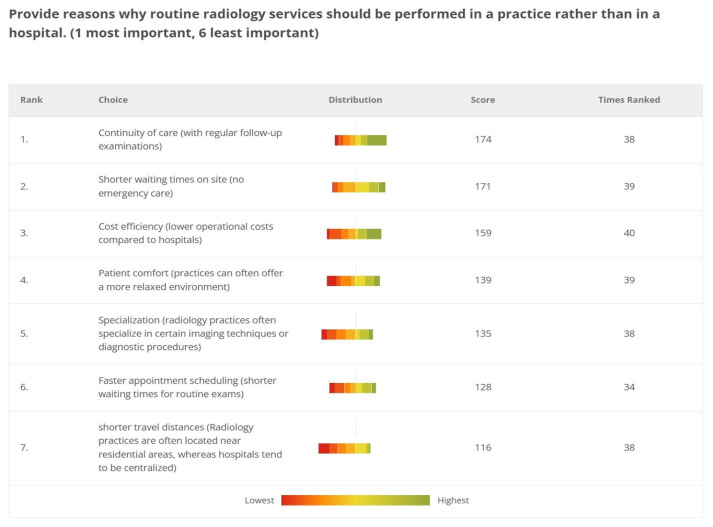
Decentralizing radiology: why private practices should handle routine imaging according to the referring physicians.

**Table 1 diagnostics-15-00962-t001:** Geographic and specialty distribution of survey participants across German federal states.

Federal State	Internists	Surgeons	General Practitioners	Radiologists	Total
Bavaria	6	5	6	4	21
North Rhine-Westphalia	7	6	6	5	24
Baden-Württemberg	5	5	4	3	17
Lower Saxony	4	3	4	2	13
Berlin	3	2	3	2	10
Hesse	3	2	4	2	11
Saxony	3	2	2	2	9
Rhineland-Palatinate	2	2	2	2	8
Schleswig-Holstein	2	2	2	1	7
Brandenburg	2	1	2	1	6
Thuringia	2	1	2	1	6
Saxony-Anhalt	1	1	2	1	5
Hamburg	1	1	2	1	5
Bremen	1	1	1	1	4
Saarland	1	1	1	0	3
Mecklenburg-Vorpommern	1	1	0	0	2
Total	50	40	48	33	171

**Table 2 diagnostics-15-00962-t002:** Specialties and experience of the participants.

	Number of Participants	Percentage	Average Experience in Years	Standard Deviation in Years	Mode in Years
Internists	50	29.24%	6.9	6.4	6
Surgeons	40	23.39%	11.5	11.24	10
General Practitioners	48	28.07%	19.2	10.6	15
Radiologists	33	19.30%	5.0	2.32	5

## Data Availability

Data are contained within the article.

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
