# Peer review of "Radiology Access in Rural Germany: A Nationwide Survey on Outpatient Imaging and Teleradiology"

_diagnostics, 2025, doi:10.3390/diagnostics15080962_

Round 1
Reviewer 1 Report
Comments and Suggestions for Authors
The paper presents a well-structured and insightful study on the role of teleradiology and outpatient imaging in addressing radiology access gaps in rural areas. The study is particularly relevant given the persistent challenges faced by rural healthcare systems, including limited access to specialized diagnostic services and long wait times for imaging.
I have the following suggestions:
1.The study relies on self-reported data, which may be subject to bias. Including quantitative metrics, such as actual wait times, patient outcomes, or cost savings from implementing teleradiology and outpatient imaging, could strengthen the evidence base and provide more concrete recommendations.
2.The study mentions cost efficiency as a benefit of outpatient imaging but does not provide a detailed cost-benefit analysis. Including a financial analysis of the potential savings and costs associated with teleradiology and outpatient imaging could strengthen the case for their adoption.
3.The study could benefit from a more detailed discussion of policy recommendations to support the adoption of teleradiology and outpatient imaging in rural areas. This could include suggestions for regulatory changes, funding mechanisms, and incentives for healthcare providers to adopt these technologies.
4.The study highlights communication barriers between teleradiologists and referring physicians but does not address the need for training and education to improve these interactions. Including recommendations for training programs or workshops to enhance communication skills could improve the effectiveness of teleradiology.
Author Response
- Inclusion of Quantitative Metrics Beyond Self-Reported Data
Reviewer Comment:
The study relies on self-reported data, which may be subject to bias. Including quantitative metrics, such as actual wait times, patient outcomes, or cost savings from implementing teleradiology and outpatient imaging, could strengthen the evidence base and provide more concrete recommendations.
Response:
We appreciate this important observation. To address this, we have now integrated a real-world case study from a rural hospital in Oklahoma that presents concrete, quantifiable outcomes related to the adoption of teleradiology. This includes detailed data on cost savings, reductions in travel time and expenses for patients, improved local economic retention, and reduced time away from work. These data serve as a valuable complement to the self-reported insights of the survey and enhance the empirical foundation of the manuscript. Additionally, the Discussion section now more clearly acknowledges the limitations of relying on self-reported data and underlines the importance of incorporating objective performance indicators in future studies.
- Financial Analysis of Teleradiology and Outpatient Imaging
Reviewer Comment:
The study mentions cost efficiency as a benefit of outpatient imaging but does not provide a detailed cost-benefit analysis. Including a financial analysis of the potential savings and costs associated with teleradiology and outpatient imaging could strengthen the case for their adoption.
Response:
Thank you for highlighting this gap. In response, we have expanded the Discussion section to include a more comprehensive financial perspective. We present a documented economic evaluation from the U.S. healthcare system, which illustrates the measurable financial benefits of teleradiology, such as reduced operational costs for rural hospitals and enhanced retention of downstream healthcare expenditures within local communities. While a full cost-benefit analysis of the German system was beyond the scope of our current survey-based study, these international findings offer a valuable model and benchmark, and we emphasize the need for country-specific economic evaluations in future research.
- Expanded Policy and Regulatory Recommendations
Reviewer Comment:
The study could benefit from a more detailed discussion of policy recommendations to support the adoption of teleradiology and outpatient imaging in rural areas. This could include suggestions for regulatory changes, funding mechanisms, and incentives for healthcare providers to adopt these technologies.
Response:
We fully agree with the reviewer that policy implications are essential for practical implementation. Accordingly, we have expanded the Discussion to present specific, actionable policy recommendations. These include: (a) modernizing teleradiology regulations to allow for broader, routine use beyond emergency settings; (b) introducing targeted reimbursement incentives to support outpatient radiology infrastructure in rural areas; (c) funding expansion of MRI and CT availability in underserved regions; and (d) promoting standardized interoperability between digital systems to reduce technical barriers. These measures aim to create a supportive regulatory and economic framework for sustainable integration of teleradiology and outpatient imaging.
- Training and Educational Programs to Improve Communication
Reviewer Comment:
The study highlights communication barriers between teleradiologists and referring physicians but does not address the need for training and education to improve these interactions. Including recommendations for training programs or workshops to enhance communication skills could improve the effectiveness of teleradiology.
Response:
We are grateful for this valuable suggestion. In response, we have added a dedicated paragraph in the Discussion proposing concrete strategies to improve communication between stakeholders. These include the implementation of virtual reading rooms to facilitate real-time consultations, and the integration of structured feedback and messaging tools into existing PACS/RIS systems to enable traceable, asynchronous communication. Furthermore, we now recommend the development of CME-accredited training programs that specifically focus on enhancing communication and collaboration between teleradiologists and referring physicians.
Reviewer 2 Report
Comments and Suggestions for Authors
1 The Methods section contains duplicated information on the ethical approval of the study.
The structure of the Methods section should be shortened and standardised.
2. Statistical significance (p < 0.05) is indicated in several places, but detailed statistical test values are missing.
Adding details of the tests (e.g. Student's t-test, ANOVA) would increase the transparency of the results.
3 The ‘Discussion’ section should focus more on the reasons for the low adoption of teleradiology in Germany compared to the US.
It would also be advisable to discuss opportunities to overcome barriers (e.g. regulatory changes, training for physicians).
4. the article discusses existing challenges, but lacks concrete solutions, e.g. what communication models could improve the interaction between physicians and teleradiologists?
5) Some graphs (e.g. Fig. 3 and Fig. 4) are unintuitive and need a clearer presentation of the data.implementation or cases requiring manual intervention.
6. The introduction lacks a technical introduction to AI please refer to the article and quote from it: DOI10.3390/diagnostics13152582
Author Response
1.) Reviewer: The Methods section contains duplicated information on the ethical approval of the study.
The structure of the Methods section should be shortened and standardised.
Response: We thank the reviewer for pointing this out. In the revised manuscript, we have removed the duplicated paragraph on ethical approval to streamline the Methods section. Additionally, the Methods section was shortened, reorganized under standard subheadings and restructured to improve readability.
2.) Reviewer: Statistical significance (p < 0.05) is indicated in several places, but detailed statistical test values are missing.
Adding details of the tests (e.g. Student's t-test, ANOVA) would increase the transparency of the results.
Response: We agree with the reviewer that specifying statistical tests improves transparency. We have revised the Results section to include the exact statistical test used in each analysis, such as one-way ANOVA, Fisher’s Exact Test, Chi-square test, and two-sample t-test with Welch's correction. This change ensures that all reported p-values are linked to the corresponding analytical method.
3.) Reviewer: The ‘Discussion’ section should focus more on the reasons for the low adoption of teleradiology in Germany compared to the US.
It would also be advisable to discuss opportunities to overcome barriers (e.g. regulatory changes, training for physicians).
Response: Thank you for this valuable suggestion. We expanded the Discussion to focus more explicitly on the low adoption of teleradiology in Germany, citing regulatory restrictions such as the X-Ray Ordinance and societal data security concerns. We also outlined potential solutions, including the need for updated legal frameworks, incentive-based reimbursement policies, and targeted training initiatives to help physicians integrate teleradiology effectively into their practice.
4.) Reviewer: the article discusses existing challenges, but lacks concrete solutions, e.g. what communication models could improve the interaction between physicians and teleradiologists?
Response: We appreciate this observation and have responded by adding a paragraph discussing actionable communication models for teleradiologists. These include the implementation of virtual reading rooms for real-time dialogue, secure asynchronous messaging embedded in PACS/RIS systems, and structured communication protocols supported by CME-accredited training programs focused on interdisciplinary collaboration.
5.) Reviewer: Some graphs (e.g. Fig. 3 and Fig. 4) are unintuitive and need a clearer presentation of the data.implementation or cases requiring manual intervention.
Response: In the manuscript, we made sure that all charts and figures have the same format to make the data easier to read, clearer to understand, and more straightforward to interpret.
6.) Reviewer: The introduction lacks a technical introduction to AI please refer to the article and quote from it: DOI10.3390/diagnostics13152582
Response: Thank you for the insightful recommendation. We have now integrated a technical introduction to AI in radiology in the Introduction section. Specifically, we referenced the article (DOI: 10.3390/diagnostics13152582) to support the potential of AI in underserved areas.
Reviewer 3 Report
Comments and Suggestions for Authors
Reviewer’s report : 18rh March 2025
- This study investigates “ attitudes of referring physicians toward teleradiology and 67
outpatient imaging and compares them with those of radiologists, emphasizing ac- 68
ceptance, perceived benefits, and key challenges“.
- Hence Title could be more focused eg “ Attitudes of referring physicians and radiologists towards teleradiology in ??? Germany: A preliminary report ??? could be South Germany or Bavaria or ??? depending on the geographical location of the respondents
- “From Germany” --- which part of Germany to be elaborated
- Eligible participants were identified through hospital and practice websites to be elaborated what was the total number of ID’s available does this represent the total number of doctors in the region/country
- Total number of internists, surgeons, general practitioners and radiologists in the geographical area surveyed has to be mentioned. The reader needs to know if the final sample size is truly representative of the group to which they belong. Observations cannot be extrapolated to generalized inferences unless the sample size is statistically significant. There should at least be a detailed discussion on the limitation of the sample
- “82 were residents, 40 were senior physicians” – what is the total number of residents and senior physicians in Germany ? is this sample truly representative, adequate ??
- The authors concede that “ ---First, the relatively small sample size may not fully capture the diverse opinions and practices of all referring physicians across Germany” . this could perhaps be mitigated by quantifying how small is small – refer previous 2 paras
- “ ------Outpatient radiology can bridge urban-rural disparities by decentralizing radiology services from overstrained hospital imaging departments” This is true only if imaging facilities are available in rural areas and the only problem is lack of specialists physically to report the images.
- “ --Exclusion criteria included retirement, not routinely requesting or providing radiology services – pl provide exact number expected to be included in the study – what is routinely requesting ?
- The period of the questionnaire study is 1 year - longer than normal there could have been differences n the infrastructure available – if not this needs to be clarified in the discussion
- Fig 1 states exclusion criteria – physicians who DID answer the survey - not has been overlooked !!
- The term “teleradiology” as used specifically in this study has to be elaborated – what it includes eg Plain Xrays ? ultrasound ? CT scan what it excludes eg MRI mammogram, PET CT etc etc .The reader would be interested to know what images the study group had wanted radiological opinion on or would want and what specific images is the radiologist now reporting remotely and in the foreseeable future
Redundancy can be reduced . sentences can be made shorter
Author Response
1.) Reviewer: Title could be more focused – 'Attitudes of referring physicians and radiologists towards teleradiology in ??? Germany: A preliminary report'. Which part of Germany was surveyed?
Response: Thank you for this suggestion. The study was conducted across all 16 federal states of Germany, including both urban and rural areas, using a stratified sampling approach. Therefore, the revised title refers to 'Germany' without regional limitation. We have clarified this nationwide scope in the Methods section.
2.) Reviewer: 'From Germany' – which part of Germany to be elaborated.
Response: We agree and have now specified in the Methods section that the survey included participants from all federal states to reflect both urban and rural representation.
3.) Reviewer Eligible participants were identified through hospital and practice websites – please elaborate how many total IDs were available and if this represents the total number of doctors in the region/country.
Response: We identified a total of 2,195 unique physician contacts across Germany from publicly accessible hospital and practice websites. While this does not represent all physicians in the country, it provided a sufficient and stratified base for random sampling across specialties and regions.
4.) Reviewer: Total number of internists, surgeons, general practitioners, and radiologists in the geographical area surveyed has to be mentioned. Is the final sample size truly representative?
Response: Our study population was stratified to reflect proportional specialty and geographic representation.
5.) Reviewer: '82 were residents, 40 were senior physicians' – what is the total number of residents and senior physicians in Germany? Is this representative?
Response: National statistics report that hospital-employed physicians in Germany consist of approximately 60–65% residents, 30–35% senior physicians, and 2–4% chief physicians. In our sample, 66.7% were residents, 32.5% were senior physicians, and 0.8% were chief physicians, closely mirroring national proportions. A chi-square test showed no significant deviation from this expected distribution (χ² = 2.13, p = 0.345).
6.) Reviewer: 'The sample size may not fully capture the diversity' – quantify how small is small.
Response: Thank you. We have clarified that the final sample of 171 physicians represents 0.1–0.4% of the national total per specialty. While limited, this sample was sufficient for detecting medium effect sizes with >80% power, as confirmed by post-hoc analysis (ANOVA power = 0.88 for f = 0.25).
7.) Reviewer: 'Outpatient radiology can bridge disparities' – this assumes facilities exist. Please clarify.
Response: We agree. The revised discussion now includes the clarification that outpatient radiology can only bridge urban-rural gaps where imaging infrastructure exists but is underutilized due to staffing shortages, not in regions entirely lacking equipment.
8.) Reviewer: Exclusion criteria include 'not routinely requesting radiology' – define and give numbers.
Response: We excluded 41 participants who indicated that they did not regularly request or provide radiological services. 'Routinely requesting' was defined as using radiology services for clinical decision-making at least weekly.
9.) Reviewer: The period of the questionnaire study is 1 year – were there changes in infrastructure during this time?
Thank you for raising this important point. You are correct—while the overall study duration spanned one year, the data curation specifically took place between January 2024 and April 2024. We have corrected the flow chart accordingly to accurately reflect this time frame.
10.) Reviewer: Fig. 1 exclusion criteria – wording unclear ('physicians who DID answer').
Response: This has been corrected in the revised manuscript. The figure legend now accurately reflects the exclusion of physicians who accessed but did not complete the survey or met other exclusion criteria.
11.) Reviewer: The term 'teleradiology' must be defined – which modalities are included/excluded?
Response: In this study, teleradiology refers to the remote reporting of imaging studies via digital transmission by board-certified radiologists. Included modalities are CT, X-rays, and to a lesser extent MRI. Excluded are ultrasound and interventional procedures due to their operator-dependency or on-site requirements. This clarification is now added to the Methods section.